# High-Fat Diet Aggravates Cerebral Infarct, Hemorrhagic Transformation and Neuroinflammation in a Mouse Stroke Model

**DOI:** 10.3390/ijms22094571

**Published:** 2021-04-27

**Authors:** Coline Grisotto, Janice Taïlé, Cynthia Planesse, Nicolas Diotel, Marie-Paule Gonthier, Olivier Meilhac, David Couret

**Affiliations:** 1Diabète Athérothrombose Thérapies Réunion Océan Indien, INSERM, UMR 1188, Université de La Réunion, 2 rue Maxime Rivière, 97400 Sainte-Clotilde, La Réunion, France; coline.grisotto@chu-reunion.fr (C.G.); janice.taile@univ-reunion.fr (J.T.); cynthia.planesse@univ-reunion.fr (C.P.); nicolas.diotel@univ-reunion.fr (N.D.); marie-paule.gonthier@univ-reunion.fr (M.-P.G.); olivier.meilhac@inserm.fr (O.M.); 2CHU de la Réunion, Service de Neuroréanimation, 97410 Saint-Pierre de la Réunion, La Réunion, France; 3CHU de la Réunion, 97400 Saint-Denis de la Réunion, La Réunion, France

**Keywords:** ischemic stroke, hemorrhagic transformation, neuroinflammation, diet-induced chronic hyperglycemia, impaired glucose tolerance, metabolic disorder

## Abstract

Background: Stroke in context of type 2 diabetes (T2D) is associated with a poorer outcome than in non-diabetic conditions. We aimed at creating a new reproducible mouse model of stroke in impaired glucose tolerance conditions induced by high-fat diet. Methods: Adult C57BL6 mice were fed for 2 months with either normal diet (ND) or high-fat diet (HFD). We used a model of Middle Cerebral Artery Occlusion (MCAO) for 90 min. Oral Glucose Tolerance Test (OGTT) and Insulin Tolerance Test (ITT) were used to assess pre-diabetic status. Brain infarct volume, hemorrhagic transformation (HT) as well as systemic and cerebral inflammatory markers were evaluated. Results: HFD was associated with an increased body weight and glycemia following OGTT. The HFD group presented a significant increase in brain infarct volume (38.7 (IQR 30–46.7%) vs. 28.45 (IQR 21–30%); *p* = 0.016) and HT (HFD: 2 (IQR 1–5) vs. ND: 0 (IQR 0–1); *p* = 0.012) and higher levels of IL-6 and MCP-1 in infarcted hemisphere compared to the ND group. Conclusion: Two months of HFD in adult mice were sufficient to alter the lipid profile and the control of hyperglycemia. These metabolic perturbations were significantly associated with increased infarct volume and hemorrhagic complications.

## 1. Introduction

Stroke is the first cause of long-term disabilities in developed countries and is also the fourth leading cause of death in the United States [1]. The main risk factors for stroke are hypertension, tobacco exposure, diabetes, dyslipidemia, poor quality diet, physical inactivity, and obesity [2]. Approximately 80% of strokes correspond to ischemic strokes which are caused by the obstruction of a cerebral blood vessel, resulting in brain cell necrosis [3]. In such cases, intravenous fibrinolytic therapy combined with endovascular thrombectomy is widely used to achieve rapid revascularization, and to improve survival and functional outcome [4,5,6,7]. Despite an increase in the frequency of symptomatic intracerebral hemorrhage (ICH), this treatment significantly improves the clinical outcome [8,9]. However, the benefits of these therapies may be partially or totally negated by cerebral ischemic reperfusion injury (CIRI). Infarct size and clinical severity at admission represent the best predictors of short-term functional outcome [10]. In a large prospective observational cohort study of early thrombolysis, massive hemorrhage was also reported to be a predictor of poor outcome [11]. The probability of poor outcome is proportional to the extent of hemorrhage which is significantly associated with history of diabetes mellitus, higher blood glucose levels and higher systolic blood pressure at baseline [12]. Chronical inflammation and vascular oxidative stress induced by obesity and chronic hyperglycemia contribute to blood–brain barrier (BBB) disruption and leakage, subsequently increasing ischemia and HT [13,14,15]. 

The incidence of obesity in industrialized countries has sharply increased and is now reaching epidemic proportions. Obesity is a health issue that leads to increased morbidity and mortality from cardiovascular diseases including stroke [16,17]. Beyond the evidence of a relationship between obesity and ischemic disease, efforts have been made to determine how obesity may affect disease severity, using animal models of stroke [18,19]. Obesity is considered as an inflammatory and prothrombotic condition with high levels of pro-inflammatory cytokines [20]. Most preclinical studies used rodents genetically deficient in leptin or leptin receptor, targeting this appetite-regulating adipokine (*ob/ob* and *db/db* mice, or *fa/fa* Zucker rat) [21]. Recently, data have shown that the detrimental effects of obesity on stroke could also be observed in models of diet-induced obesity (DIO) [22]. DIO appears as more clinically relevant because mutations leading to impaired leptin signaling are only found in a small subset of obese humans [23]. The deleterious effects of DIO on stroke in mice are not fully understood and characterized. Obesity can alter the local inflammatory response observed during a stroke, contributing to worsening CIRI and leading to poor neurological outcome [24]. The aims of this study were to determine the effects of a two-month HFD protocol on peripheral parameters including body weight, glucose and insulin tolerance and, above all, to study its potential effects in a 90-min brain ischemia using the MCAO model with a particular focus on neuroinflammation, infarct volume and HT.

## 2. Results

### 2.1. High-Fat Diet Induces Weight Gain and Impairs Glucose Tolerance

Seven-week-old mice were subjected to 8 weeks of HFD and several parameters were investigated such as body weight, insulin resistance (insulin tolerance test: ITT), as well as response to glucose by an oral glucose tolerance test (OGTT). At the end of the 8-week HFD protocol, the body weight mean was 26.4 (IQR 22–32.7 g) vs. 24.8 (IQR 21.4–28.2 g) in HFD and controls (normal diet: ND), respectively (*n* = 24–28; *p* = 0.03, Figure 1A). Moreover, although an 8-week HFD protocol did not induce insulin resistance as shown by the ITT (Figure 1B), it altered the ability to regulate blood glucose levels in response to oral gavage (OGTT, Figure 1C), or to conditions of stress due to surgery (Figure 1D). Interestingly, HFD mice had higher hyperglycemia after surgical incision (before MCAO) than ND mice (13.1 ± 0.4 mmol/L vs. 11.5 ± 0.4 mmol/L *p* = 0.03) and up to 1 h after MCAO. Taken together, these results showed that the 8-week HFD protocol developed in this study was sufficient to induce a significant weight gain and an early pre-diabetic state characterized by glucose level dysregulation without insulin resistance. 

### 2.2. HFD Alters Metabolic and Inflammatory Plasma Markers

In naive mice, the 8-week HFD protocol significantly increased plasma triglyceride levels compared to control mice (ND SHAM 0.106 ± 0.013 mmol/L vs. HFD SHAM 0.151 ± 0.025 mmol/L *p* = 0.028 and ND MCAO 0.112 ± 0.008 mmol/L vs. HFD MCAO 0.152 ± 0.01 mmol/L *p* = 0.006) (Figure 2A). Likewise, triglyceride levels were positively correlated to the body weight (Figure 2B). Similarly, plasma cholesterol levels were also significantly increased after HFD (ND SHAM 1.9 ± 0.2 mmol/L vs. HFD SHAM 3.1 ± 0.6 mmol/L *p =* 0.03 and ND MCAO 2.1 ± 0.44 mmol/L vs. HFD MCAO 3.6 ± 0.4 mmol/L *p* = 0.005) and significantly correlated with body weight (Figure 2C,D). Furthermore, HFD also induced a significant upregulation of plasma leptin levels compared to control mice, with leptinemia correlating with body weight independently of the MCAO condition (ND SHAM 143 ± 26.3 pg/mL vs. HFD SHAM 407 ± 54.8 pg/mL *p* < 0.001) (Figure 2E,F). These results show that an 8-week HFD altered the plasma lipid profile and leptinemia, probably due to an increase in the amount of adipose tissue, both being correlated with body weight. Given that overweight/obesity is associated with chronic inflammation, plasma levels of a major pro-inflammatory cytokine IL-6 were measured [25]. Interestingly, in these experimental conditions, the HFD protocol failed to modify significantly plasma IL-6 levels (ND SHAM 9.78 ± 4.9 pg/mL vs. HFD SHAM 12.99 ± 10 pg/mL; *p* = 0.64) (Figure 3). However, stroke resulted in a systemic pro-inflammatory state regardless of diet, as shown in Figure 3 (ND SHAM 9.78 ± 4.9 pg/mL vs. ND MCAO 37.6 ± 9.7 pg/mL *p* = 0.02 and HFD SHAM 12.99 ± 10 pg/mL vs. HFD MCAO 40.95 ± 13.5 pg/mL *p* = 0.03). Altogether, these data show that HFD and related weight gain increased triglyceridemia, cholesterolemia and leptinemia but not plasma IL-6 levels. In addition, stroke per se produced an acute systemic pro-inflammatory state reflected by diet-independent increase in plasma IL-6 levels. 

### 2.3. Impaired Glucose Tolerance Induced by HFD Is Associated with Increased Brain Damage in a 90-Min MCAO Model 

In order to investigate the potential impact of impaired glucose tolerance during stroke, ND and HFD mice were subjected to 90 min of cerebral ischemia using the MCAO method. The infarct size was subsequently quantified after TTC staining 24 h after stroke, and the HT was assessed. In the HFD group, the ischemic volume was significantly increased compared to that of ND (38.7 (IQR 30–46.7%) vs. 28.45 (IQR 21–30%), respectively; *p* = 0.016) (Figure 4A,B). Additionally, the HT score was assessed (Figure 4C). HFD resulted in an increased HT score relative to that of controls (2 (IQR 1–5) vs. 0 (IQR 0–1), for HFD and ND, respectively; *p* = 0.012) (Figure 4D). However, at 24 h from stroke onset, neurological outcome scores were not significantly different between both groups (HFD: 3.05 ± 0.33 vs. ND: 2.33 ± 0.45 *p* = 0.56). Taken together, these results showed that two months of HFD were sufficient to aggravate brain damage and CIRI without producing a diabetic state. 

### 2.4. HFD Promotes Local Brain Inflammation

We investigated the neuroinflammatory parameters in the brain of MCAO mice, focusing on two pro-inflammatory markers, namely IL-6 and MCP-1. In ND conditions, acute stroke increases IL-6 levels in the ischemic hemisphere compared to the contralateral hemisphere (97.05 (IQR 40.8–147) vs. 27 (IQR 20.8–30.8) ng/g; *p* = 0.039) (Figure 5A). Moreover, in HFD conditions, a more significant increase in IL-6 levels was observed in the ischemic hemisphere compared to the contralateral hemisphere (134 (IQR 60–220) vs. 41.6 (IQR 25.4–54.2) ng/g; *p* < 0.0001) (Figure 5B). Of note, HFD led to a higher increase in IL-6 levels in ischemic hemisphere than that measured in ND condition (134 (IQR 60–220) vs. 97.05 (IQR 40.8–147) ng/g; *p* = 0.019) (Figure 5C). The cerebral production of MCP-1 was also more elevated in ischemic vs. contralateral hemispheres under both ND and HFD conditions (Figure 5D,E) (ND: ischemic: 157 (IQR 130–205) ng/g vs. contralateral: 83 (IQR 71.5–124.5) ng/g; *p* = 0.003 and HFD: ischemic: 248 (IQR 171.5–284.5) ng/g vs. contralateral: 98 (IQR 68–148.5) ng/g; *p* < 0.0001) (Figure 5E). Furthermore, MCP-1 levels were significantly higher in the ischemic hemisphere of HFD mice compared to ND ones (248 (IQR 171.5–284.5) vs. 157 (IQR 130–205) ng/g; *p* = 0.023) (Figure 5F). Altogether, these results confirmed a worsening in local neuroinflammation induced by HFD during acute cerebral ischemia.

## 3. Discussion

In clinical studies, acute hyperglycemia is associated with an increased risk of in-hospital mortality after ischemic stroke, of intra-cerebral symptomatic hemorrhage (regardless of rtPA treatment), and with a risk of poor functional recovery in stroke survivors [26,27]. The most commonly used model of ischemic stroke with hyperglycemia is based on the destruction of pancreatic beta cells by streptozotocin [28]. However, this model does not reproduce the clinical situation observed in humans. We have already demonstrated that an intraperitoneal injection of glucose before MCAO induced acute hyperglycemia and led to an aggravation of ischemic and hemorrhagic lesions [29]. It is a reproducible model for therapeutic preclinical studies on hemorrhagic complications during acute reperfusion of stroke. Mice used in this study were young and healthy. Acute hyperglycemia has a detrimental effect on cerebral reperfusion lesions. In our experimental conditions, we observed a stress hyperglycemia during acute ischemic stroke as demonstrated in clinical study [27]. In the present study, only 2 months of HFD led to a more pronounced surgical stress hyperglycemia than in ND condition. The detrimental effect, particularly HT, on the ischemic brain after only 2 months of exposure to HFD is the novelty of this article. Indeed, the time frame of exposure to HFD increasing HT during the acute phase of stroke remains unclear in the literature. Many studies have shown the negative impact of long exposure to HFD beyond 3 months on brain damage [18,30]. A recent study suggested that a very acute and short 3-day HFD exposure worsens ischemic damage during AIC but not HT [31]. This reproducible model may help us better understand the pathophysiology and timing of these complications during prediabetes states. Moreover, our results show that only 2 months of HFD in healthy adult C57BL6 mice induced impaired glucose tolerance without insulin resistance or true diabetic status. Insulin resistance precedes type 2 diabetes and occurs when insulin signaling is impaired. In this pathological condition, pancreatic beta cells normally produce more insulin to maintain glucose homeostasis. Over time, the pancreas is no longer able to support adequate insulin secretion, leading to the development of impaired glucose tolerance and to an established diabetic condition [32]. In our experimental conditions, the insulin response was not altered suggesting a prediabetes status.

We also found an increase in plasma leptin, triglycerides and cholesterol in the HFD group. Plasma leptin levels are greatly correlated with adipose tissue [33]. This is an accurate recapitulation of the clinical situation, as individual patients are rarely solely obese, but often present with other co-morbidities, including hypertension, diabetes and dyslipidemias. In this study, our model greatly reproduced the metabolic syndrome. During obesity, the adipose tissue constitutes a major site of low-grade inflammation, leading to insulin resistance. Indeed, gut microbiota dysbiosis induced by a HFD leads to gut permeability (altered distribution of tight junction proteins), which in turn promotes metabolic endotoxemia and initiates the development of low-grade inflammation and insulin resistance in the liver, muscles and adipose tissue [34,35]. In mice, a study showed that nutritional circulating fatty acids activate toll-like receptor 4 (TLR4) signaling in adipocytes and macrophages, and demonstrated the capacity of fatty acids to induce inflammatory signaling in adipose cells or tissue [36]. It is well established that this pro-inflammatory mediator interferes with insulin signaling in peripheral tissues through activation of c-JUN N-terminal kinase (JNK) and nuclear factor-kappa B (NF-kB) pathways [37,38]. In summary, HFD induces low-grade systemic inflammation by increasing endotoxin levels (e.g., LPS), circulating free fatty acids and inflammation mediators whose homeostasis has been altered in many organs [15,39]. Interestingly, our results show that stroke per se leads to systemic inflammation characterized by a significant increase in plasma levels of IL-6 in MCAO groups independently of the diet conditions as compared to SHAM groups. This acute systemic inflammatory response to brain damage has already been described in preclinical studies. The absence of increase in IL-6 plasma levels according to the diet condition is probably due to the preferential tissue production of this cytokine. In fact, the main part of IL-6 secreted by the adipose tissue is then taken up by the liver to induce the subsequent production of C-reactive protein inflammatory marker.

Stroke triggers a peripheral immune response by both humoral and neural routes, with release of lymphocytes from immune organs, increased inflammatory mediators in several organs, and sustained increase in other circulating markers of inflammation [40,41]. 

In our study, only 2 months of HFD were sufficient to induce more cerebral ischemic damage during MCAO than in ND. Indeed, we found a significant increase in ischemic volume in the HFD group. These results are consistent with clinical studies reporting that patients with post-stroke hyperglycemia have poorer outcomes than those with normal blood glucose levels [27]. Several mechanisms have been proposed to explain the deleterious effects of hyperglycemia on ischemic brain tissue. Acute hyperglycemia was shown to reduce endothelium vasodilation by inhibition of endothelium-derived nitric oxide synthesis, which leads to a reduction of cerebral blood flow and decreases reperfusion in the ischemic tissue. Moreover, hyperglycemia worsens reperfusion injury by increasing oxidative stress and inflammation [42]. We also found a significant increase in HT in the HFD group. The reason why 2-month-old HFD mice are more susceptible to HT during acute stroke is unknown. In clinical setting, HT is associated with infarct volume [43]. Our 2-month HFD mouse model are enough to increase cerebral damage and infarct size that could aggravate HT. We could also hypothesize that the HFD promotes chronic inflammation in the brain and favors BBB disruption. Taïlé et al. showed that hyperglycemic conditions increased the permeability to FITC-Dextran and decreased the production of occludin, claudin-5, ZO-1 and ZO-2 tight junctions in cerebral endothelial cells [44]. In mice model of stroke, Arcambal et al. found that VE-Cadherin (a specific endothelial protein) levels were reduced under hyperglycemic condition, leading to impaired BBB integrity [45]. In our study, we found an increase in MCP-1 expression in the ischemic hemisphere, even more pronounced in the HDF group. This result could be due to the presence of blood cells in the brain during HT. MCP-1 could increase leukocyte infiltration in the ischemic brain. Leukocytes, mainly neutrophils, are an important source of MMP-9 that contributes to early disruption of the BBB in ischemic stroke [46,47]. In humans, MMP-9-positive neutrophils accumulate in infarcted and hemorrhagic areas and high levels of MMP-9 have also been associated with BBB breakdown and hemorrhagic complications [48]. Moreover, hyperglycemia contributes to greater ICH by an increased stimulation of MMP-9 activity [49]. However, due to the presence of other comorbidities such as dyslipidemia, obesity and pre-diabetic states, it becomes difficult to attribute the worse poor outcome in this model as being solely due to hyperglycemia. Human clinical trials concluded that treatment with intravenous insulin to maintain normoglycemia after ischemic stroke did not provide any benefit in terms of functional outcomes, and exposed, more frequently, patients to acute hypoglycemia [50,51]. Obesity is an independent risk factor for stroke and is associated with poor outcome after stroke. The present study allows us to hypothesize that one of the mechanisms of the aggravation of brain damage by obesity could be poor glycemic control associated with a pre-diabetic condition and a low-grade inflammation.

In conclusion, our model demonstrates a significant association between impaired glucose tolerance and stroke severity with increased ischemic volume and hemorrhagic complications. This result is of particular interest in a model of pre-diabetic mice, without insulin resistance. This mouse model of stroke in pre-diabetic conditions could be used to test new therapeutic approaches and/or to improve the understanding of the disease.

## 4. Materials and Methods

### 4.1. Animals and Diet

All experiments were carried out in strict compliance with the protocol for the Animal Ethics Committee of CYROI Réunion Island who approved the in vivo protocol (A 974 001). The research protocol was designed in accordance with the principles of replacement, reduction and refinement fitting in with the European Directive (Directive 2010/63/EU) on the protection of animals used for scientific purposes. C57BL6 mice were purchased from Charles River Laboratory. In the HFD group, 28 C57BL6 mice of 7 weeks of age (sex ratio 1/1) were fed for 8 weeks with HFD. The HFD consisted in 43% of total calculated energy from lipids, 21% of total calculated energy from protein and 35% from carbohydrates (SF04-001 23% Fat Semi Pure Rodent Diet, Specialty Feeds). For the control group, 24 mice of 15 weeks of age (sex ratio of 1/1) were purchased from Charles River laboratory (France). They were fed with ND which contained 8.4% of lipids, 19.3% of proteins and 72.4% of carbohydrates (A04; UAR, Villemoisson-sur-Orge, France).

### 4.2. Body Weight and Glycemia Evolution

In the HFD group, food intake, body weight and capillary glycemia were measured weekly in non-fasting animals and blood was sampled from the intraorbital retrobulbar plexus in anesthetized mice. For capillary glycemia measurement, blood was sampled from the tail of conscious mice with an ACCU-CHECK© glucometer.

### 4.3. Oral Glucose Tolerance and Insulin Tolerance Tests

After 8 weeks of HFD, 12 animals were subjected to an Insulin Tolerance Test (ITT) and 13 animals to an Oral Glucose Tolerance Test (OGTT). In the control group, each test was performed on 6 mice, respectively. An ITT was conducted using Actrapid, Novolin R human insulin (Novo Nordisk^®^) at 1 UI/kg body weight, administered by subcutaneous injection. Animals were fasted (5 h), and blood glucose was tested at baseline (time = 0 min) and at 30, 60, 90 and 120 min after injection. Then, animals were fasted for 12 h and OGTT was performed using glucose (glucose 30%), administered by gavage (2 g/kg body weight). Blood glucose was measured at baseline (time = 0 min) and at 15, 30, 45, 60, 90 and 120 min after oral gavage.

### 4.4. Stroke Model

Cerebral ischemia was induced by middle cerebral artery occlusion (MCAO) achieved by intraluminal introduction of a silicon-coated monofilament into the MCA, according to the method previously published [29]. Briefly, in anesthetized mice, a surgical cervical midline incision was made to expose the common, internal and external right carotid arteries (respectively CCA, ICA and ECA). The CCA was ligated, and the ICA was temporarily closed by a clamp. A small incision was then made in the CCA and a silicon-coated monofilament (diameter of 0.21 ± 0.02 mm) was inserted into the ICA through the CCA. The filament was advanced to the anterior cerebral artery, bypassing and occluding the origin of MCA. After 90 min of occlusion, the filament was withdrawn to allow blood reperfusion. A sham surgery was performed in 9 mice in the control group and 6 mice in the HFD group. Sham mice were subjected to the same surgical procedure without MCAO.

### 4.5. Neurological Evaluation 

Functional deficit at 24 h post-reperfusion was scored using a five-point Bederson’s scale: scale 0, no deficit; 1, mild forelimb weakness; 2, severe forelimb weakness, consistently turns to side of deficit when lifted by tail; 3, compulsory circling; 4, unconscious; and 5, death [52]. 

### 4.6. Evaluation of Infarct Volume and Intracerebral Hemorrhage

#### 4.6.1. Infarct Volume Assessment

At 20 h after the reperfusion, mice were euthanized under general anesthesia by cardiac puncture. Brains were immediately removed and cut into 5 serials of 1 mm-thick coronal slices. A 2% solution of 2,3,5-triphenyltetrazolium chloride (TTC) was used to assess the infarct zone (white zone). Image-J^®^ (image-processing software) was used by an independent observer blinded to the group status to measure ischemic area (the unstained areas) and to calculate the infarct volume in each mouse brain. 

#### 4.6.2. Hemorrhagic Score Assessment

An observer blinded to the group status also scored the intracerebral hemorrhage in brain slices according to the CT scan scale used in clinical practice: (0) none, (1) small petechial hemorrhagic infarction, (2) confluent petechial hemorrhagic infarction, (3) small parenchymal hemorrhage (<30% of infarct, mild mass effect) and (4) large parenchymal hemorrhage (>30% of infarct, marked mass effect) [53]. The hemorrhagic score was obtained by adding scores of each brain slice (with a total score out of 20) [54]. There are two types of HT, early HT measured between 18 to 24 h after stroke and late HT measured beyond 24 h after stroke. In this study, we focused our experiments on early HT due to blood recirculation leading to cerebral reperfusion in all euthanized 24 h after stroke mice. According to literature data, late HT is due to increased vascular permeability and increased blood flow following the reduction of the cerebral edema [55,56,57].

### 4.7. Evaluation of Brain and Systemic Inflammation

#### 4.7.1. Enzyme-Linked Immunosorbent Assay for Interleukin-6 (IL-6) and Leptin

Levels of pro-inflammatory IL-6 and leptin were determined in plasma, by using a Mouse IL-6 ELISA kit (eBioscience^®^, ThermoFisher Scientific, Illkirch Cedex, France) and a Mouse Leptin ELISA kit (RayBiotech^®^, Norcross, GA, USA), respectively. The assays were performed according to the manufacturer’s instructions and the absorbance was measured at 450 nm (FLUOstar Optima^®^, Bmg Labtech, Offenburg, Germany).

#### 4.7.2. Enzyme-Linked Immunosorbent Assay for IL-6 and MCP-1 in Brain

Homogenate levels of infarcted and contralateral hemisphere were quantified by using IL-6 and MCP-1 ELISA kits (Abcam, Paris, France), respectively. The assays were performed according to the manufacturer’s instructions and the absorbance was measured at 450 nm (FLUOstar Optima^®^, Bmg Labtech, Offenburg, Germany). Values were normalized against total protein contents measured in ischemic or contralateral hemisphere by BCA assay.

#### 4.7.3. Colorimetric Enzymatic Assay

Total cholesterol and triglyceride levels were determined in plasma, by using standard enzymatic methods (Cholesterol FS^®^, DiaSys, Belgique and Triglyceride quantification colorimetric kit, Biovision, Nanterre, France). The absorbance was measured at 500 nm for total cholesterol and 570 nm for triglycerides, respectively (FLUOstar Optima^®^, Bmg Labtech, Offenburg, Germany).

### 4.8. Statistical Analysis

All quantitative data were expressed as mean ± standard error of the mean (SEM) or median with interquartile range and analyzed using unpaired Student’s *t*-test as indicated (Excel Office 2011^®^ for Mac). Differences with statistical significance are denoted by *p* < 0.05. The results are reported as frequency (*n*), percentage (%), mean, SEM, or median with interquartile range. Continuous variables (glucose levels, HT score, ischemic volume, composite score) were compared between groups using Mann–Whitney nonparametric tests. IL-6 and MCP-1 variables were compared between groups using Wilcoxon matched-paired signed rank test. Association between continuous and categorical variables was tested using Pearson’s correlation test. All statistical analyses were performed using GraphPad PRISM^®^ software. Differences were considered statistically significant for *p* < 0.05.

## Figures and Tables

**Figure 1 ijms-22-04571-f001:**
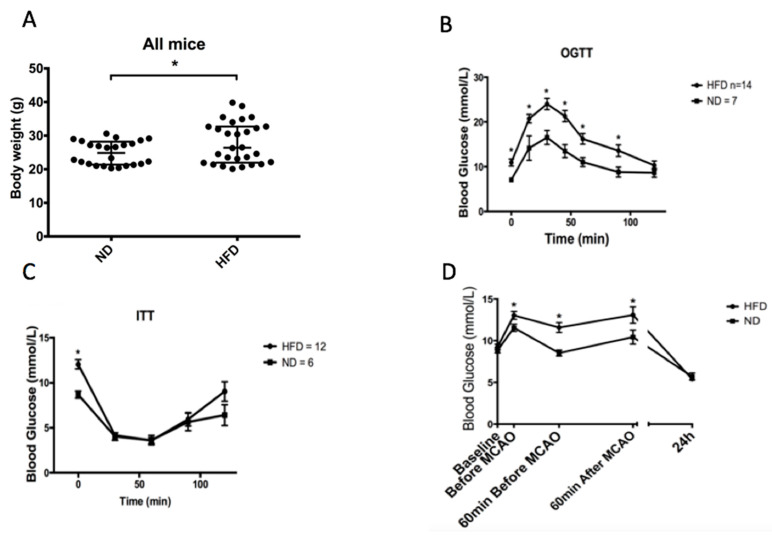
Obesity and pre-diabetic states. (**A**) Body weight, the day of the surgery in all mice. ND (*n* = 24) 24.8 (IQR 21.4–28.2 g) vs. HFD (*n* = 28) 26.4 (IQR 22–32.7 g); * *p* = 0.033; (**B**) ITT: No difference between HFD and ND was observed concerning insulin response; (**C**) OGTT: Glycemia was monitored for 100 min. HFD vs. ND groups: * *p* < 0.05; (**D**) Impact of MCAO surgery on glycemia. HFD group have a more pronounced stress hyperglycemia relative to ND group * *p* < 0.05.

**Figure 2 ijms-22-04571-f002:**
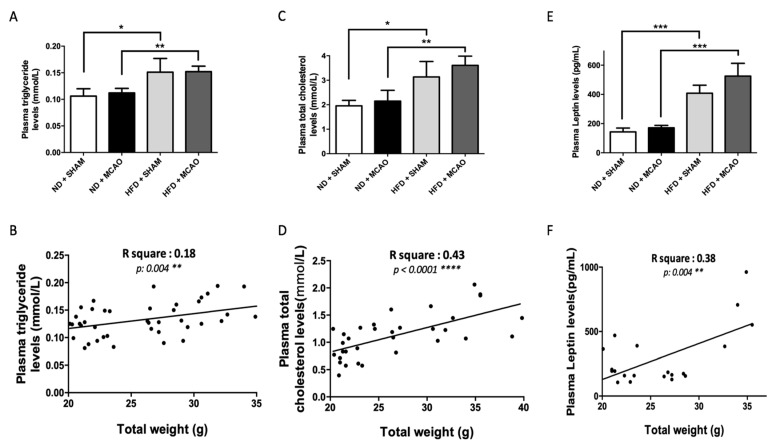
Metabolic deregulation illustrating metabolic syndrome. (**A**) Two months of HFD leads to an increase in plasma triglyceride levels. * *p* < 0.05 (ND + SHAM vs. HFD + SHAM) ** *p* < 0.001 (ND + MCAO vs. HFD + MCAO) (*n* = 3 per group); (**B**) Correlation between triglyceridemia levels and body weight; (**C**) Plasma cholesterol levels are higher in HFD groups relative to ND groups; * *p* < 0.05 (ND + SHAM vs. HFD + SHAM) ** *p* < 0.001 (ND + MCAO vs. HFD + MCAO) (*n* = 3 per group); (**D**) Correlation between cholesterolemia levels and body weight; (**E**) HFD increases significantly leptin plasma levels relative to ND; *** *p* < 0.0001 (ND + SHAM vs. HFD + SHAM and ND + MCAO vs. HFD + MCAO) (*n* = 3 per group); (**F**) Positive correlation between leptinemia and body weight.

**Figure 3 ijms-22-04571-f003:**
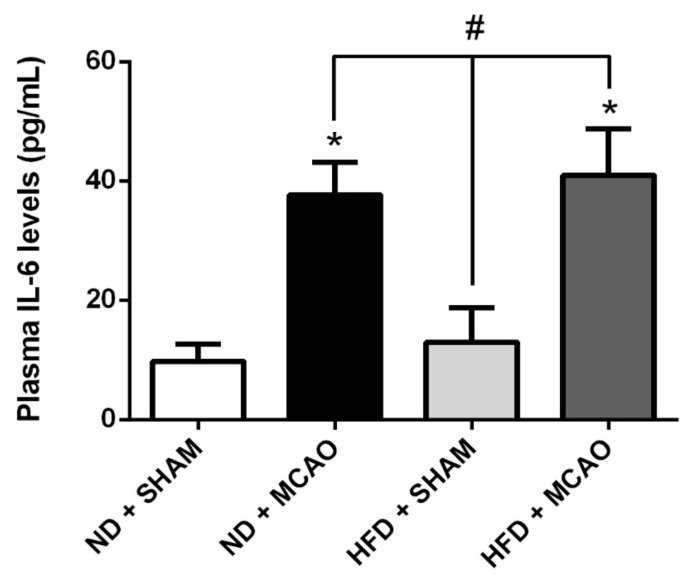
Stroke induced acute systemic inflammation. Plasma IL-6 levels: MCAO induces a significant increase in IL-6 plasma concentration whereas the HFD did not produce any effect on this parameter (*n* = 3 per group). Data are expressed as means ± SEM. *: *p* < 0.05 as compared to ND + SHAM; #: *p* < 0.05 as compared to HFD + SHAM.

**Figure 4 ijms-22-04571-f004:**
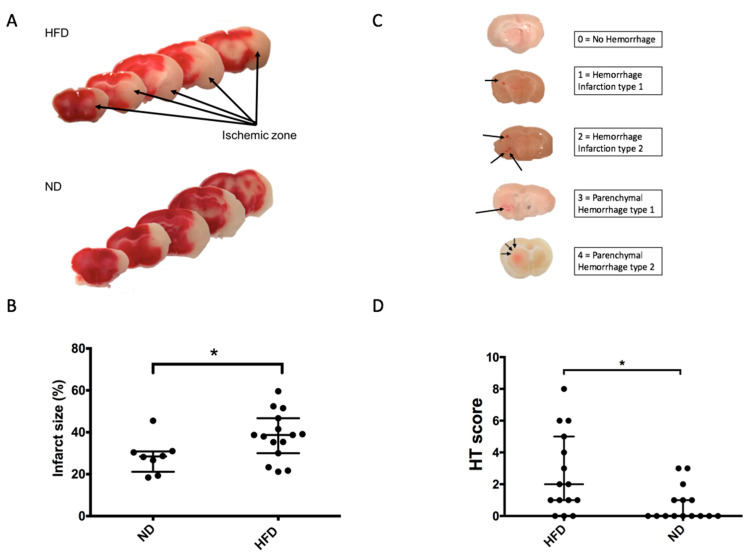
HFD promotes brain injuries during acute stroke. (**A**) TTC staining of coronal brain slice. In red, healthy tissue and in white, ischemic/necrotic zone (non-stained); (**B**) Ischemic volume in % 24 h after reperfusion (HFD: 38.7 (IQR 30–46.7%) *n* = 15 vs. ND: 28.45 (IQR 21–30%), *n* = 8); * *p* = 0.038. All value are median with (IQR); (**C**) Examples of hemorrhagic transformation for the score determination; (**D**) Hemorrhage transformation score 24 h after MCAO in coronal brain slice (total of slice individual score in 5 slices) (HFD group: 2 (IQR 1–5) *n* = 15 vs. ND 0 (IQR 0–1), *n* = 15; * *p* = 0.02). All values are median with (IQR).

**Figure 5 ijms-22-04571-f005:**
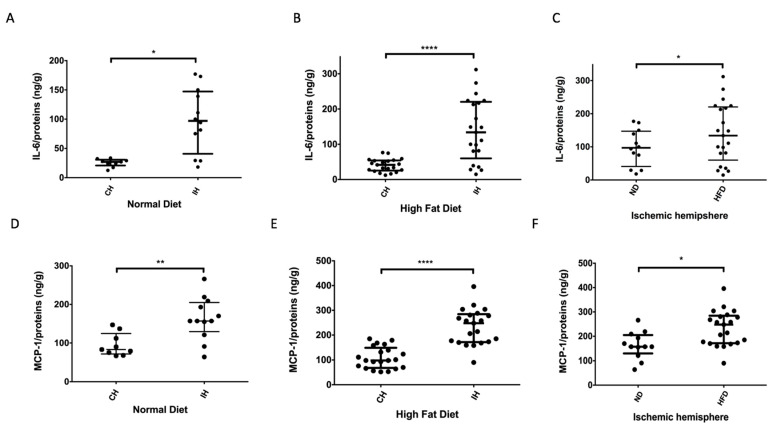
Brain IL-6 and MCP-1 quantification. (**A**) Under normal diet (ND), there is statistically significant IL-6 increase in ischemic hemisphere (IH) relative to contralateral hemisphere (CH) *n* = 12; * *p* = 0.039. (**B**) Under HFD, brain IL-6 level is increased in IH vs. CH (*n* = 21); **** *p* < 0.0001. (**C**) Significant difference was observed for IL-6 comparing IH in HFD and ND conditions, ND *n* = 12 and HFD *n* = 21; * *p* = 0.019. (**D**,**E**) Increased in brain MCP-1 level in IH vs. CH was statistically significant in ND *n* = 12; ** *p* = 0.003 and in HFD conditions, HFD *n* = 21; **** *p* < 0.0001. (**F**) The increase in MCP-1 level observed in the IH is significantly more important in the HFD group (*n* = 21) relative to the ND group (*n* = 12); * *p* = 0.023.

## Data Availability

The date that support the findings of this study are available from the corresponding author upon reasonable request.

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
