# Peer review of "High-Fat Diet Aggravates Cerebral Infarct, Hemorrhagic Transformation and Neuroinflammation in a Mouse Stroke Model"

_ijms, 2021, doi:10.3390/ijms22094571_

Round 1

Reviewer 1 Report

The paper by Grisotto et al. reports the detrimental effect of a high-fat diet (model of type-2 diabetes) in the mouse brain leading to an increased lesion size, hemorrhagic transformation and inflammation.

The strength of the paper is its relevance to a direct translational application in stroke unit, and this aspect is not extensively studied. The limitations of the paper are 1) the report of the data, and 2) the lack of discussion in view with similar data reported in a model of type-1 diabetes induced by streptozotocin in the mouse brain (Poittevin et al., Diabetes, 2015, 64, 999-1010. The methods are mostly appropriate.

Specific points to revise:

1) The most important concerns the presentation of data (see below).

Data are reported as mean+/-SEM.

For your information, read the following paragraph about how to report data in the right way.

Bar charts are not appropriate for continuous measures since they do not provide information about distribution of the data. Appropriate descriptive measures of the average and variability for continuous measures in tables, text, or graphs are the arithmetic mean and standard deviation if the data are sufficiently normally distributed, or the median and interquartile range [being the 25th and 75th percentile] if data are not sufficiently normally distributed, but not the standard error (SE). The presentation of the standard error as measure of variability is not correct since it is a 67% confidence coefficient for the mean, meaning that the interval mean ± standard error of the mean is a 67% confidence interval of the mean.

Consequently, for Figures 1A, 4B and 4D, 5 A-F provide dots and median with interquartile range. For figure 2 (A, C and E) and figure 3, and as the number of samples is small use the mean +/- SD.

2) Results

- According to the Figure 4D only 50% of animals exhibited a hemorrhagic transformation. This result should be discussed in view of the status of the microcirculation. Can this hemorrhagic transformation appear at a later time in these animals?

Author Response

Specific points to revise:

1) The most important concerns the presentation of data (see below).

Data are reported as mean+/-SEM.

For your information, read the following paragraph about how to report data in the right way.

Bar charts are not appropriate for continuous measures since they do not provide information about distribution of the data. Appropriate descriptive measures of the average and variability for continuous measures in tables, text, or graphs are the arithmetic mean and standard deviation if the data are sufficiently normally distributed, or the median and interquartile range [being the 25th and 75th percentile] if data are not sufficiently normally distributed, but not the standard error (SE). The presentation of the standard error as measure of variability is not correct since it is a 67% confidence coefficient for the mean, meaning that the interval mean ± standard error of the mean is a 67% confidence interval of the mean.

Consequently, for Figures 1A, 4B and 4D, 5 A-F provide dots and median with interquartile range. For figure 2 (A, C and E) and figure 3, and as the number of samples is small use the mean +/- SD.

Thank you for this important comment. We have made the corrections in the revised version of the manuscript.

2) Results

- According to the Figure 4D only 50% of animals exhibited a hemorrhagic transformation. This result should be discussed in view of the status of the microcirculation. Can this hemorrhagic transformation appear at a later time in these animals?

Thank you for this comment, which will allow us to clarify this point. After ischemic stroke, HT are usually dichotomized into early HT (<18-24h after stroke) and late HT (>24h after stroke). The mechanisms involved in these complications seem to be different. Classically, early HT are due to blood recirculation, leading to reperfusion, whereas late HT are due to increased vascular permeability and increased blood flow following a reduction of the cerebral edema [1-3]. Our study was designed to investigate only early HT since mice were euthanized 24h after stroke. We totally agree that HT could be underestimated by this short timing. Hemorrhagic transformation could indeed occur later in this animal model. This point is now discussed in the revised version of the manuscript: “There are two types of HT, early HT measured between 18 to 24 hours after stroke and late HT measured beyond 24 hours after stroke. In this study, we focused our experiments on early HT due to blood recirculation leading to cerebral reperfusion in all euthanized 24 hours after stroke mice. According to literature data, late HT is due to increased vascular permeability and increased blood flow following the reduction of the cerebral edema [1-3]

  1. Jickling GC, Liu D, Stamova B, Ander BP, Zhan X, Lu A, Sharp FR: Hemorrhagic transformation after ischemic stroke in animals and humans. J Cereb Blood Flow Metab 2014, 34(2):185-199.
  2. Fisher M, Adams RD: Observations on brain embolism with special reference to the mechanism of hemorrhagic infarction. J Neuropathol Exp Neurol 1951, 10(1):92-94.
  3. Hornig CR, Dorndorf W, Agnoli AL: Hemorrhagic cerebral infarction--a prospective study. Stroke 1986, 17(2):179-185.

Reviewer 2 Report

Comment 1) Did the HT had any correlation with infarction size? In this article, the authors argued that the hemorrhagic transformation was developed by cerebral ischemic reperfusion injury (CIRI). In this study, artery was reopened after 90 minutes. Thus, there may be association between HT and CIRI. However, it has been known that the extent of HT was associated with massive cerebral infarction and area of infarction (HT has been known to be occurred in gray matter, especially in the cerebral cortex, because of its abundant collateral circulation). I think that If the infarction volume was correlated with the degree of HT, as the cause of high degree of HT, I recommend to add this explanation -> large infarction volume after HFD protocol also may be the another reason of high degree of HT.

Comment 2) In figure 3, plasma IL-6 level was not significantly different between ND+MCAO and HFD+MCAO. The infarction volume in HFD+MCAO was significantly higher that ND+MACO, but not in neuroinflammatory markers. Please clarify or suggest the reason of this findings.

Comment 3) In page 9, line 316, “The filament was advanced to the anterior carotid artery….”. What is anterior carotid artery?

Comment 4) Please add the reference for hemorrhagic score assessment

Comment 5) page 8, line 269, “Stroke did not provide any benefit in terms of functional outcome and exposed more frequently~” Please add this most recent article (Karen et al. JAMA Neurology 2019;322:326-335) about the benefit of glucose control in acute ischemic stroke

Comment 6) Page 8 line 265, worse outcome may be replaced with poor outcome.

Author Response

Comment 1) Did the HT had any correlation with infarction size? In this article, the authors argued that the hemorrhagic transformation was developed by cerebral ischemic reperfusion injury (CIRI). In this study, artery was reopened after 90 minutes. Thus, there may be association between HT and CIRI. However, it has been known that the extent of HT was associated with massive cerebral infarction and area of infarction (HT has been known to be occurred in gray matter, especially in the cerebral cortex, because of its abundant collateral circulation). I think that If the infarction volume was correlated with the degree of HT, as the cause of high degree of HT, I recommend to add this explanation -> large infarction volume after HFD protocol also may be the another reason of high degree of HT.

We fully agree with the Reviewer #2 that another reason for the higher HT prevalence observed in the HFD group could be the larger infarct. In order to improve our article, we added this explanation in the revised manuscript: “. In clinical setting, HT is associated with infarct volume [4]. Our 2-month HFD mouse model are enough to increase cerebral damage and infarct size that could aggravate HT

  1. Bozzao L, Angeloni U, Bastianello S, Fantozzi LM, Pierallini A, Fieschi C: Early angiographic and CT findings in patients with hemorrhagic infarction in the distribution of the middle cerebral artery. AJNR Am J Neuroradiol 1991,12(6):1115-1121.

Comment 2) In figure 3, plasma IL-6 level was not significantly different between ND+MCAO and HFD+MCAO. The infarction volume in HFD+MCAO was significantly higher that ND+MACO, but not in neuroinflammatory markers. Please clarify or suggest the reason of this findings.

As judiciously pointed out by the Reviewer #2, in our study, we did not find an increase in plasma IL-6 according to the diet condition 24h after MCAO. This result is in line with our previous data showing that MCAO may be mainly responsible for the increase in plasma IL-6 levels. Indeed, in Arcambal et al. article [5], we found that MCAO alone was sufficient to induce a 3-fold increase in plasma IL-6 level. A combination of MCAO with acute hyperglycemia did not exacerbate this up-regulation of plasma IL-6 concentration. Moreover, here the inability of the HFD+MCAO condition to potentiate the elevation of plasma IL-6 mediated by ND+MCAO may result from the absence of a pronounced HFD action on plasma IL-6 levels. Changes in IL-6 production are more expected at the tissue levels, as we reported here in the brain. It is well known that HFD-induced obesity leads to an increased local production of IL-6 from the adipose tissue, keeping in mind that the main part of IL-6 secreted by the adipose tissue is then taken up by the liver to induce the subsequent production of C-reactive protein inflammatory marker. We added this sentence to clarify this point: The absence of increase in IL-6 plasma levels according to the diet condition is probably due to the preferential tissue production of this cytokine. In fact, the main part of IL-6 secreted by the adipose tissue is then taken up by the liver to induce the subsequent production of C-reactive protein inflammatory marker”

  1. Arcambal A, Taile J, Couret D, Planesse C, Veeren B, Diotel N, Gauvin-Bialecki A, Meilhac O, Gonthier MP: Protective Effects of Antioxidant Polyphenols against Hyperglycemia-Mediated Alterations in Cerebral Endothelial Cells and a Mouse Stroke Model. Mol Nutr Food Res 2020, 64(13):e1900779.

Comment 3) In page 9, line 316, “The filament was advanced to the anterior carotid artery….”. What is anterior carotid artery?

We apologize for this mistake. In fact, the correct sentence is “anterior cerebral artery”. The correction has been made in the revised manuscript.

Comment 4) Please add the reference for hemorrhagic score assessment

Thank you for this comment, two references have been added in the revised manuscript [6, 7].

  1. Fiorelli M, Bastianello S, von Kummer R, del Zoppo GJ, Larrue V, Lesaffre E, Ringleb AP, Lorenzano S, Manelfe C, Bozzao L: Hemorrhagic transformation within 36 hours of a cerebral infarct: relationships with early clinical deterioration and 3-month outcome in the European Cooperative Acute Stroke Study I (ECASS I) cohort. Stroke 1999, 30(11):2280-2284.
  2. Lapergue B, Dang BQ, Desilles JP, Ortiz-Munoz G, Delbosc S, Loyau S, Louedec L, Couraud PO, Mazighi M, Michel JB et al: High-density lipoprotein-based therapy reduces the hemorrhagic complications associated with tissue plasminogen activator treatment in experimental stroke. Stroke; a journal of cerebral circulation 2013, 44(3):699-707.

Comment 5) page 8, line 269, “Stroke did not provide any benefit in terms of functional outcome and exposed more frequently~” Please add this most recent article (Karen et al. JAMA Neurology 2019;322:326-335) about the benefit of glucose control in acute ischemic stroke

Thank you for this suggestion. This reference has been added in the revised manuscript.

Comment 6) Page 8 line 265, worse outcome may be replaced with poor outcome.

Thank you for this appropriate suggestion. The correction has been made.

Reviewer 3 Report

To date, it has been sufficiently shown that hyperglycemia exacerbated stroke pathogenesis, yielding increases in infarct area and hemorrhagic transformation. It is rationale that inflammatory cytokines mediate such mechanisms. What is the novelty of this study? Also, Introduction is too long.

Author Response

Reviewer #3

To date, it has been sufficiently shown that hyperglycemia exacerbated stroke pathogenesis, yielding increases in infarct area and hemorrhagic transformation. It is rationale that inflammatory cytokines mediate such mechanisms. What is the novelty of this study? Also, Introduction is too long.

The aim of our study was not to show that hyperglycemia increased the risk of hemorrhagic transformation, but rather to provide a new model of pre-diabetic mice induced by a high-fat diet allowing the scientific/medical community to test various therapies. This model mimics not only chronic hyperglycemia but also metabolic perturbations in response to high-fat diet. We agree with the reviewer that our objectives were not clearly exposed and we have thus modified accordingly the manuscript in order to underline the relevance of our model from a patho-physiological point of view. The novelty of our study relies on the reproducibility of the HFD-induced pre-diabetic state that can be achieved in only 8 weeks of feeding, which is sufficient to reproduce HT, similar to those observed in clinical practice in DT2 patients. This sentence has been added to the revision manuscript: “The detrimental effect, particularly HT, on the ischemic brain after only 2 months of exposure to HFD is the novelty of this article. Indeed, the time frame of exposure to HFD increasing HT during the acute phase of stroke remains unclear in the literature. Many studies have shown the negative impact of long exposure to HFD beyond 3 months on brain damage [8, 9]. A recent study suggested that a very acute and short 3-day HFD exposure worsens ischemic damage during AIC but not HT [10]. This reproducible model may help us better understand the pathophysiology and timing of these complications during prediabetes states”

  1. Maysami S, Haley MJ, Gorenkova N, Krishnan S, McColl BW, Lawrence CB: Prolonged diet-induced obesity in mice modifies the inflammatory response and leads to worse outcome after stroke. J Neuroinflammation 2015, 12:140.
  2. Langdon KD, Clarke J, Corbett D: Long-term exposure to high fat diet is bad for your brain: exacerbation of focal ischemic brain injury. Neuroscience 2011, 182:82-87.
  3. Haley MJ, Krishnan S, Burrows D, de Hoog L, Thakrar J, Schiessl I, Allan SM, Lawrence CB: Acute high-fat feeding leads to disruptions in glucose homeostasis and worsens stroke outcome. J Cereb Blood Flow Metab 2019, 39(6):1026-1037.

We agree with the Reviewer #3 that the introduction was too long. This section has been reduced in order to improve the quality of our manuscript.

Round 2

Reviewer 1 Report

The authors have made revisions, but it seems that only partial revisions have been done.

Are the authors sure that on Fig. 1A and 4 A and B, it is represented the median with interquartile range (IQR), because up and down variabilities appear similar (in each specific group), whereas in the text of the legend it is indicated (as an exemple in Fig. 4A: 38.7 [30-46.7]%)??

Please make corrections in the two figures.

Author Response

Reviewer #1 R2

The authors have made revisions, but it seems that only partial revisions have been done.

Are the authors sure that on Fig. 1A and 4 A and B, it is represented the median with interquartile range (IQR), because up and down variabilities appear similar (in each specific group), whereas in the text of the legend it is indicated (as an exemple in Fig. 4A: 38.7 [30-46.7]%)??

Please make corrections in the two figures.

Thank for this comment. We have made the corrections in the revised version of our manuscript

Reviewer 3 Report

Manuscript was improved. I have no more comments.

Author Response

Thank you